# Minima-YOLO: A Lightweight Identification Method for Lithium Mineral Components Under a Microscope Based on YOLOv8

**DOI:** 10.3390/s25072048

**Published:** 2025-03-25

**Authors:** Zeyang Qiu, Xueyu Huang, Xiangyu Xu

**Affiliations:** 1School of Software Engineering, Jiangxi University of Science and Technology, Nanchang 330013, China; qq2462218642@163.com (Z.Q.); xuxiangyu@jxust.edu.cn (X.X.); 2School of Electronic Information Industry, Jiangxi University of Science and Technology, Ganzhou 341600, China

**Keywords:** mineral detection, lepidolite, edge device, YOLOv8, lightweight network

## Abstract

Mineral identification technology is a critical technology in the construction of smart mines. To enable effective deployment and implementation of rapid mineral sorting for valuable ores on edge computing devices, we propose a lightweight identification method for lithium minerals under visible light microscopy based on YOLOv8, named Minima-YOLO. First, by scaling down and limiting the number of channels in the YOLOv8 backbone, we introduced a smaller network, YOLOv8-tiny. Next, we redesigned a new lightweight feature extraction module, Faster-EMA, using PConv and the EMA attention mechanism, replacing the original C2f module. Third, we incorporated GhostConv, a cost-effective downsampling method, as a replacement for standard convolutions. Finally, to mitigate the impact of deeper backbone network layers, we introduced the Slim-Neck structure in the Neck, further reducing the model size. Ultimately, Minima-YOLO achieves a 99.4% mAP50 on our self-constructed lithium mineral image dataset, with FLOPs reduced to 2.3 G, parameters to 0.72 M, and a model size of just 1.63 MB, while maintaining an FPS of 103. A series of comparative experiments confirmed its superior performance over other advanced object detection algorithms. This algorithm, with its highly efficient lightweight design and rapid inference speed, provides an intelligent, efficient, and eco-friendly computer vision method for the rapid sorting of lithium mineral components.

## 1. Introduction

Lepidolite is the most common lithium mineral and is also the key raw material for lithium extraction in the modern new energy industry. As a critical element in the 21st century, lithium plays an irreplaceable role in the energy revolution. With the rapid development of electric vehicles, energy storage batteries, and consumer electronics [1], the International Energy Agency (IEA) estimates that the global demand for lithium has grown at an annual rate of over 25% over the past decade [2]. By 2030, global lithium consumption is expected to exceed 2 million tons, more than three times the current production. This supply-demand contradiction makes the efficient development and precise identification of lithium resources an important issue in national resource security strategies.

Conventional mineral detection methodologies predominantly rely on physicochemical spectroscopic analysis [3] and chemical quantification techniques [4]. These approaches typically necessitate manual execution of multistep operational protocols, demonstrating inherent limitations in procedural complexity, time inefficiency, and labor intensity, which collectively impede their applicability to modern industrial requirements for large-scale deployment and real-time monitoring. The evolution of computer vision technology has catalyzed a paradigm shift in mineralogical identification, establishing vision-based algorithmic frameworks as a critical research frontier in mineral detection [5].

Despite the considerable potential of computer vision in mineral identification, significant technical challenges remain to be addressed across various mineral categories and application environments [6]. For instance, Zhang et al. [7] systematically identified critical limitations in current automated mineral recognition technologies, including the scarcity of publicly accessible databases, challenges in multimodal learning frameworks, and deployment constraints of portable sensing systems. Specifically, the morphological similarities between certain mineral species may induce feature ambiguity in computer vision systems, compromising classification accuracy. Concurrently, variations in illumination conditions, heterogeneous image quality, and complex mineral textures further degrade the robustness and reliability of recognition algorithms. The misclassification of economically valuable minerals, leading to their erroneous disposal into tailings, may result in substantial financial losses [8,9]. Furthermore, the limited computational capacity of edge computing devices in industrial settings, compared to conventional computing systems, imposes significant constraints on the execution of complex models, potentially hindering deployment efficiency or introducing operational latency [10].

To address the aforementioned challenges, we propose a highly lightweight detection model by optimizing the network architecture and integrating low-cost convolutional techniques with attention mechanisms. The proposed model is designed to achieve rapid and accurate identification of lithium mineral composition while maintaining sufficient lightweight characteristics for deployment on edge devices. Ultimately, it enables fast identification, classification, and purity evaluation of refined minerals in industrial applications, providing valuable reference and guidance for practical production.

The main contributions and methodologies of this study are as follows:A lithium ore block image dataset was independently constructed through field sampling, imaging, processing, and other preparatory steps.We proposed the YOLOv8-tiny network. Based on YOLOv8n, we proportionally reduced the number of channels in the backbone and further limited the max channels. This significantly reduces the computational load while maintaining the overall architecture’s ratio, making the network more lightweight and compact.We redesigned a new lightweight bottleneck structure by introducing partial convolution and EMA attention mechanisms. Based on that, we further proposed a new feature extraction module (Faster-EMA) to replace the C2f module in the backbone of YOLOv8-tiny, reducing computational load while maintaining strong feature extraction capability.We introduced Ghost Convolution and a Slim-Neck structure into the backbone and neck of the network, respectively. By utilizing low-cost computations and a lightweight neck design, computational overhead and network size are significantly reduced.

The structure of the remainder of this paper is as follows: Section 2 provides a review of related research on lithium mineral identification. Section 3 introduces an improved lithium mineral detection model and elaborates on the specific improvements made. Section 4 describes the dataset, experimental environment, and parameter settings, and outlines the evaluation criteria for the experiments. Section 5 analyzes the experimental results and presents the data and conclusions from various comparative experiments. Section 6 summarizes the findings of the study and discusses potential future research directions.

## 2. Related Work

Traditional lithium mineral identification methods involve manual analysis based on the physical and chemical properties of lithium minerals. For example, Pöllmann et al. [11] combined X-ray diffraction (XRD) with quantitative mineralogy and the Rietveld method to improve lithium ore evaluation methodologies. Wang et al. [12] treated samples with a nitric acid-hydrofluoric acid digestion system and accurately measured lithium content using ICP-AES. Although these methods allow for accurate detection of lithium minerals, they often require complex sample preparation and high-precision equipment, leading to low efficiency and high costs. In addition, wet chemical analysis is sensitive to environmental factors and reagent selection, often causing chemical pollution [13]. As a result, these methods are better suited for detailed analysis of small laboratory samples but fail to meet the demands of large-scale, rapid identification in modern industrial production.

With the development of machine vision, identification algorithms applied in computer image processing have partially addressed challenges that traditional methods struggle to overcome. For instance, Patel et al. [14] developed a machine vision system based on the support vector regression (SVR) algorithm to predict iron ore grade online. The system achieved a coefficient of determination (R^2^) of 0.94, validating its performance. Luo et al. [15] proposed a weighted K-nearest neighbor (K-NN) algorithm to classify zinc ore flotation conditions based on froth image features, significantly improving classification accuracy. However, traditional machine vision classifiers typically depend on manually designed feature extraction approaches, which are heavily dependent on prior expertise. These methods must be adjusted or re-engineered for different mineral identification tasks and are vulnerable to environmental changes.

To overcome these limitations, deep learning techniques have been gradually applied in industrial fields, including mineral identification [16]. Zhang et al. [17] proposed an improved deep convolutional neural network algorithm and subsequently introduced a more effective lightweight network [18] to enhance the accuracy and efficiency of surface defect detection in industrial products, with its superior performance validated through experiments. For industrial applications, Zhang and his colleagues [19] proposed a data-driven optimization method using a neural network-based identifier to reduce computational costs and enhance decision-making in complex industrial processes. As demonstrated by Xu et al. [20], who designed a U-Net convolutional neural network model based on the TensorFlow framework, this model successfully achieved intelligent recognition and classification of minerals such as pyrite and chalcopyrite, with an accuracy rate exceeding 90%. Zhou [21] and colleagues developed a sensor that utilizes deep ensemble learning to monitor and adjust conditions in real time during the froth flotation process, optimizing mineral recovery rates.

In single-stage detection algorithms, YOLO has drawn our attention due to its fast detection and high accuracy, which have been extensively applied in the detection of other minerals. For example, Tang et al. [22] proposed an improved YOLOv5-based mineral object identification algorithm; by introducing a lightweight backbone network and optimizing convolutional structures, the parameter count was reduced by 24.95%, while the mAP reached 95.7%, meeting the identification demands of industrial production. Zeng [23] and his colleagues developed a YOLOv8-enhanced method for distinguishing coal and gangue, one of the types achieving a 95.3% mAP with FLOPs of 29.7 and a size of 22.1 MB. Even so, existing YOLO models still have some limitations. YOLO is primarily designed for generalized multi-class scenarios, resulting in complex network structures with high parameter counts and computational demands. In contrast, mineral recognition tasks typically involve only a few classes, with relatively small variations in target size. As a result, the original model retains some redundancy, and there remains room for optimization tailored to specific environments.

In summary, while deep learning methods are widely used in mineral identification, and partial solutions have been developed for various challenges related to different minerals and environmental conditions, research on the rapid identification of lithium minerals remains limited. Therefore, we aimed to advance this work.

## 3. Methods

### 3.1. The Original YOLOv8 Network

YOLOv8 [24], an evolutionary advancement of the canonical YOLOv5 architecture [25], preserves the core CSP and PAN topological frameworks (Figure 1).

The key architectural innovations encompass the following: (i) substitution of C3 modules with computationally optimized C2f counterparts in backbone/neck components, (ii) implementation of decoupled detection heads replacing legacy coupled configurations, and (iii) adoption of anchor-free prediction paradigms. The enhanced architecture integrates multi-branch residual connectivity, substantially improving detection performance metrics. However, this architectural sophistication introduces elevated parameter dimensionality and computational overhead. These constraints critically impede real-time operational viability in industrial settings and exacerbate deployment limitations on edge computing platforms with stringent resource budgets.

### 3.2. The Improved Minima-YOLO Network

Considering the operational requirements and resource constraints inherent to micrometric mineral particulate analysis, a lightweight detection framework has been developed for microscale mineralogical target identification. As illustrated in Figure 2, the proposed architecture comprises three functionally specialized components: backbone, neck, and head.

The backbone extracts features from the input image, with the Faster-EMA module serving as a lightweight feature extractor. This module performs iterative operations at different stages. Additionally, the efficient Ghost Convolution module is used for downsampling.

In the neck section, feature maps of different scales are fused to produce output feature maps with resolutions of 80 × 80, 40 × 40, and 20 × 20, enhancing the model’s ability to identify mineral blocks at varying scales. The C2f module without shortcut connections is used for feature extraction. GSConv is introduced for downsampling and combined with VoV-GSCSP to construct a Slim-Neck structure in the neck.

The head consists of three decoupled heads, each responsible for calculating the regression and classification losses for feature maps at three different scales. The regression loss includes CIoU Loss (Complete IoU Loss) and DFL Loss (Distribution Focal Loss), while the classification loss is represented by Cls loss (Classification Loss).

CIoU Loss considers IoU, center point distance, and aspect ratio consistency, further enhancing the performance of boundary box regression. The specific calculation formula is as follows:(1)$$L_{CIoU}=1-IoU+\frac{\rho^{2}\left(b,b_{gt}\right)}{d^{2}}+\alpha v$$
Here, IoU is the Intersection over Union of the predicted and ground truth boxes. $\rho \left(b,b_{gt}\right)$ is the Euclidean distance between their centers. d is the diagonal length of the smallest enclosing box. α is a weight, and v is a correction factor, defined as(2)$$\alpha =\frac{v}{(1-IoU)+v}$$(3)$$v=\frac{4}{\pi^{2}}{\left(arctan\frac{w_{gt}}{h_{gt}}-arctan\frac{w}{h}\right)}^{2}$$
w and h represent the width and height of the predicted box, while $w_{gt}$ and $h_{gt}$ are those of the ground truth box. αv jointly evaluates the aspect ratio consistency between the predicted and ground truth boxes.

DFL Loss focuses on modeling the discretization of bounding box coordinates and optimizes predictions through distribution weights. The specific formula is as follows:(4)$$L_{DFL}=\frac{1}{N}\sum_{i=1}^{N}\;\sum_{k=1}^{K}\;w_{k}\cdot log\left({\overset{ˆ}{p}}_{k}\right)$$ Here, K is the number of discrete positions, ${\overset{ˆ}{p}}_{k}$ is the predicted probability of a discrete value (after Softmax), and $w_{k}$ denotes the distance between the target value and the discrete position:(5)$$w_{k}=\left|x_{\mathrm{target}}-x_{k}\right|$$
$x_{\mathrm{target}}$ is the target value, and $x_{k}$ is the position of the discrete value.

Cls loss evaluates the difference between the predicted class probabilities and the ground truth classes. The calculation formula is as follows:(6)$$L_{Cls}=-\frac{1}{N}\sum_{i=1}^{N}\;\sum_{c=1}^{C}\;y_{i,c}log\left({\overset{ˆ}{p}}_{i,c}\right)$$ Here, N is the total number of samples, C is the total number of classes, $y_{i,c}$ is the ground truth label, and ${\overset{ˆ}{p}}_{i,c}$ is the predicted probability by the model.

### 3.3. YOLOv8-Tiny

Given that this study focuses on a single microscopic environment with only a few classes and minimal scale variations, a relatively small-scale network is adequate to achieve the required accuracy while also ensuring faster processing speeds. Nevertheless, the smallest model in the YOLOv8 series (i.e., YOLOv8n) still exhibits some redundancy. Therefore, to better cater to our specific task, we introduced YOLOv8-tiny. While maintaining the global depth, width, and ratios, we restricted the maximum number of channels to 512 and further scaled down the number of channels in each backbone layer (layers 0 to 9) by a factor of 1/2. This makes YOLOv8-tiny slimmer and more compact compared to all baseline networks of YOLOv8. Table 1 presents a detailed comparison of channel numbers from layer 0 to layer 9, the maximum number of channels (Max_c), and the parameter quantities (Prams) among YOLOv8n, YOLOv8s, and YOLOv8-tiny.

### 3.4. Faster-EMA Module

#### 3.4.1. Partial Convolution

The Darknet Bottleneck [26] serves as a core component in YOLOv5′s C3 and YOLOv8′s C2f modules, as illustrated in Figure 3a. While residual connections improve feature propagation during training, its standard convolution layers incur significant computational overhead. To resolve this, Chen et al. [27] introduced partial convolution (PConv), which selectively processes input channels to reduce FLOPs while optimizing memory access (Figure 4). The method achieves higher computational throughput (FLOPS) than standard convolution, with FLOPs and memory access quantified as follows:(7)$$FLOPs=h\times w\times k^{2}\times c_{p}^{2}$$(8)$$Memory\;access=h\times w{\;\times \;2c}_{p}+k^{2}\;\times \;c_{p}^{2}\approx h\;\times \;w{\;\times \;2c}_{p}$$
Here, h and w represent the width and height of the feature map, k denotes the kernel size, and $c_{p}$ is the number of channels processed by standard convolution in PConv. When the proportion $c_{p}/c=1/4$, the remaining $\left({c-c}_{p}\right)$ channels do not participate in convolution, eliminating the need for memory access.

Consequently, PConv achieves memory access operations at 1/4 and FLOPs at 1/16 of standard convolution values, demonstrating substantially reduced computational complexity. Building upon this advancement, the FasterNet Block architecture was subsequently developed, integrating optimized partial convolution layers for enhanced operational efficiency, as shown in Figure 3b.

#### 3.4.2. EMA Attention Mechanism

The dual objectives of developing a computationally efficient architecture under resource-constrained conditions while preserving robust feature extraction capacity are addressed through the integration of FasterNet Block components with an Efficient Multi-Scale Attention (EMA) mechanism [28]. The structure of this module is shown in Figure 5. For any input feature map $X\in R^{c\times h\times w}$, EMA divides X into g subfeatures (groups) along the channel dimension, where each group has c/g channels. These subfeatures are then processed in parallel sub-networks.

The first two branches use 1 × 1 operations with global average pooling along the X and Y directions, while the third branch applies 3 × 3 convolution. The results of the former are concatenated, processed with a 1 × 1 convolution, and activated with Sigmoid to generate attention weights. The formula for global average pooling is as follows:(9)$$E_{c}=-\frac{1}{h\times w}\sum_{i=1}^{h}\;\sum_{j=1}^{w}\;X_{c}(i,j)$$
Here, $E_{c}\in R^{c/g}$ represents the pooled feature vector, and $X_{c}(i,j)$ denotes the feature value of the c-th channel at pixel (i, j).

The third branch extracts local spatial features using 3 × 3 convolution, and the resulting weights re-weight the features from the first two branches. Group Norm, global average pooling, and Softmax are then applied to compute attention weights, with the c-th channel’s attention weight calculated using Softmax as follows:(10)$$a_{c}=\frac{exp(E_{c})}{\sum_{k=1}^{c/g}\;exp(E_{k})}$$ Here, $E_{k}$ denotes the globally pooled feature value of the k-th channel.

The key step in cross-space learning is performing matrix multiplication between the attention distribution $a_{c}$ and the input feature map $X_{c}$. This facilitates interaction across groups and further refines the features. Finally, the feature maps of each group are output through residual connections and re-weighted using the Sigmoid function. The results are integrated across multiple attention weights, ensuring feature integrity and consistent dimensions.

The EMA module synthetically integrates multi-branch grouping mechanisms, localized convolutional operations, and global attention pathways. This architecture effectively mitigates feature degradation inherent in cross-channel interaction modeling via channel compression, a limitation prevalent in the SE module [29]. Comparative evaluations with established attention mechanisms (e.g., CBAM and ECA [30,31]) show that EMA has superior computational efficiency and is highly suitable for lightweight networks.

#### 3.4.3. Faster-EMA Bottleneck

The Faster-EMA Bottleneck was engineered through systematic integration of partial convolution and EMA mechanisms, with its comprehensive structure detailed in Figure 3c. The architecture initiates feature processing through a 3 × 3 PConv layer, synergistically integrated with convolutional, batch normalization, and activation operations for primary feature extraction.

This configuration achieves dual optimization: PConv minimizes computational complexity and memory access frequency, while the Convolution-BatchNorm-SiLu (CBS) module enhances feature discriminability without architectural over-parameterization. Subsequently, 1 × 1 convolutional operations execute dimensional reduction while establishing inter-channel correlations. The hierarchical processing culminates in the EMA module, which extracts high-order semantic features through multi-scale attention pathways.

#### 3.4.4. The Working Principle of the Faster-EMA Module

The C2f module surpasses C3 in optimizing feature propagation while preserving spatial details with a reduced computational load, which is particularly advantageous for object detection. In YOLOv8-tiny’s C2f architecture, the Darknet Bottleneck is replaced by the Faster-EMA Bottleneck to construct an enhanced feature extraction module (Figure 6).

Adhering to C2f’s bifurcation principle, input features from the backbone are split post initial convolution: one branch connects directly to the concatenation layer, while the other traverses multiple Faster-EMA Bottlenecks. Each bottleneck sequentially applies 3 × 3 PConv, CBS operations, 1 × 1 convolution, and EMA attention for hierarchical feature refinement. The final output is generated by concatenating both pathways followed by terminal convolution. Engineered as a lightweight feature extraction architecture for microscopic mineral particle identification, this module preserves the architectural merits of C2f while integrating enhanced bottleneck configurations. Utilizing partial convolution operators and an EMA mechanism, the design maintains robust feature discriminability with substantially reduced computational complexity. Such computational efficiency proves critical for real-time processing and deployment on resource-constrained edge devices.

### 3.5. Ghost Convolution

In lightweight network architectures, downsampling operations impose substantial computational burdens due to redundant feature map generation in conventional convolutional layers. To mitigate this inefficiency, Han et al. [32] developed Ghost Convolution (GhostConv), a parameter-efficient convolution method employing sparse standard convolutions to generate base features followed by linear transformations (cheap operation) for channel expansion. The process initiates with adaptive kernel standard convolutions, subsequently enhances features through computationally inexpensive operations, and culminates in channel-wise concatenation of base and transformed features (Figure 7).

The theoretical computational efficiency of GhostConv is rigorously examined herein. The output feature maps can be viewed as “ghosts” of intrinsic feature maps, which are generated through inexpensive transformations denoted by the operator ω. Here, $\omega_{i,j}$ represents the linear operation used to generate the j-th ghost feature $y_{ij}(\forall i=1,\ldots ,m,j=1,\ldots ,n)$. The final $\omega_{m,n}$ is an identity mapping, which preserves the intrinsic feature maps. Ultimately, we can obtain m×n feature maps $Y=\left[y_{11},y_{12},\ldots ,y_{mn}\right]$, which represent the output data of the GhostConv. Assuming the input feature map size is $h_{0}\times w_{0}$, with c channels, output feature map size h×w, convolution kernel size k, and m×n output channels, the computation for standard convolution can be simplified as follows:(11)$$C_{S}=h\times w\times c\times k^{2}\times (m\times n)$$

To ensure consistent spatial dimensions, the kernel size, stride, and padding in GhostConv match those of standard convolution. The first step generates m feature maps, and each map produces (n−1) new maps in the second step. Each linear operation uses an average kernel size of d. By the second step, a total of m×(n−1) feature maps are obtained. The total computation is as follows:(12)$$C_{G}=C_{first}+C_{cheap}=h\times w\times c\times k^{2}\times m+h\times w\times d^{2}\times m\times (n-1)$$
Given $k^{2}\sim d^{2}$ and c≫n, the computational cost ratio of GhostConv to standard convolution can be approximately expressed as follows:(13)$$\frac{C_{G}}{C_{S}}=\frac{h\times w\times c\times k^{2}\times m+h\times w\times d^{2}\times m\times (n-1)}{h\times w\times c\times k^{2}\times (m\times n)}=\frac{c+n-1}{c\times n}\approx \frac{1}{n}$$
The results show that Ghost Convolution requires only 1/n of the computation of standard convolution. Thus, it effectively reduces the model’s computational cost.

### 3.6. Slim-Neck

#### 3.6.1. GSConvolution

To optimize the network’s neck component, we introduced the Slim-Neck structure [33]. It incorporates GSConvolution (GSConv) to enhance both feature extraction efficiency and information retention, as illustrated in its convolution process in Figure 8.

The detailed procedure is mathematically represented as follows. For an input $X\in R^{c_{1}\times h\times w}$, the standard convolution branch is given by the following:(14)$$X_{SC}=W_{SC}^{3\times 3}*X,X_{SC}\in R^{\frac{c_{2}}{2}\times h\times w}$$
Here, "∗" denotes the convolution operation and $c_{1}$ and $c_{2}$ represent the number of input and output channels, respectively.

The depthwise separable convolution branch is given by the following:(15)$$X_{DSC}=W_{DSC}^{3\times 3}⊙X,X_{DSC}\in R^{\frac{c_{2}}{2}\times h\times w}$$
Here, “⊙” represents the depthwise separable convolution operation.

The outputs $X_{SC}$ and $X_{DSC}$ are concatenated and mixed through a Shuffle operation, which cleverly combines the strengths of channel-dense convolution and channel-sparse convolution. This approach retains hidden connections between channels while significantly reducing the computational complexity. The final optimized feature map ensures uniform feature fusion and comprehensive information integration. The GSConv process can be summarized as follows:(16)$$X_{GSConv}=GSConv(X)=Shuffle(Concat(X_{SC},X_{DSC})),X_{GSConv}\in R^{C_{2}\times h\times w}$$

#### 3.6.2. VoV-GSCSP

The GSBottleneck further constructed based on GSConv is shown in Figure 9a. It takes feature maps as the input and applies two cascaded GSConv operations in the main branch and a 3 × 3 convolution in the residual branch. The final output is produced through residual connections. The GSBottleneck operation is defined as follows:(17)$$GSBottleneck(X)={GSConv}_{2}({GSConv}_{1}(X))+W_{res}^{1\times 1}*X$$ Here, “+” denotes the residual connections in the result.

VoV-GSCSP constitutes a critical component within the Slim-Neck structure. By introducing GSBottleneck and a one-stage aggregation strategy, it effectively integrates multi-level features through hierarchical fusion of input and extracted representations. The simplified VoV-GSCSP structure is schematically illustrated in Figure 9b.

For the input feature map $X\in R^{c_{1}\times h\times w}$, the main branch performs channel reduction using a 1 × 1 convolution, followed by feature optimization through the GSBottleneck. The bypass branch retains part of the original input information and processes it with a simple 1 × 1 convolution. Finally, the main and bypass branches are concatenated and further optimized with a final convolution to produce the output feature map. The complete process of VoV-GSCSP can be expressed mathematically as follows:(18)$$X_{VoV-GSCSP}=W_{final}^{1\times 1}*Concat\left(GSBottleneck\left(W_{main}^{1\times 1}*X\right),W_{bypass}^{1\times 1}*X\right),X_{VoV-GSCSP}\in R^{C_{2}\times h\times w}$$

In practical applications, the VoV-GSCSP module improves the efficiency of feature fusion and processing, making it ideal for resource-limited edge computing.

## 4. Data and Experimental Preparation

### 4.1. Data Collection and Preprocessing

The lithium mineral image dataset was acquired from the mineral processing facility of Yongxing Special Steel New Energy Technology Co., Ltd. (Yichun, China), comprising powdered samples of feldspar, quartz, and lepidolite. Imaging was performed using a KO-series microscope (Model: KOMD0745; LED illumination: 10 W; Shenzhen, China) configured with the following operational parameters: 3840 × 2160 pixel resolution, 120 mm working distance, and a 0.5× objective lens (focal scale: 0.7; magnification: 200×). Figure 10 shows a selection of the initial lithium mineral micrographs.

Given the indistinct feature delineation among mineral particulates in preliminary micrographs, an iterative refinement process was implemented. Initial processing employed an automated segmentation protocol using segment coding [34] to isolate mineral phases while preserving the original morphological integrity. To mitigate sub-resolution debris and enhance morphological consistency, an edge-trimming and padding algorithm [35] was subsequently implemented, discarding particulates with dimensions <100 or >400 pixels and standardizing all samples to 640 × 640 pixel resolution via zero padding. The procedural workflow is detailed in Figure 11.

A lithium mineral dataset comprising 3,201 annotated instances (feldspar: 1071; quartz: 1099; lepidolite: 1031) was constructed through this methodology. To ensure morphological and environmental diversity, samples were selected across varied morphological configurations (e.g., differential lighting intensities and particulate aggregation states), as exemplified in Figure 12. Post-processing enhancements achieved enhanced textural granularity and inter-mineral discriminability, as follows: (i) feldspar: opaque morphology with specular reflections under high-intensity illumination; (ii) quartz: translucent structure exhibiting vitreous luster and thickened edges; (iii) lepidolite: lamellar transparency with minimal reflectance and reduced edge thickness.

The dataset, annotated via LabelImg [36] with YOLO-compliant “txt” classifications, was partitioned into training/validation/testing subsets (8:1:1 ratio), as detailed in Table 2. This standardized framework provides a robust foundation for machine vision-driven mineralogical classification and feature quantification.

### 4.2. Dataset Augmentation

This study employed YOLO’s Mosaic data augmentation (originally introduced in YOLOv4 [37]) to enhance the model generalization capacity and mineralogical recognition accuracy. This technique stochastically selects geometric anchor points ($x_{c},y_{c}$) to synthesize quad-image composites, resizing and aligning four distinct micrographs into a unified training sample while adaptively recalibrating bounding boxes for detection consistency (Figure 13). This approach amplifies textural granularity and spatial feature representation critical for mineral identification, outperforming conventional augmentation methods (rotation/scaling/color jitter) through enhanced data diversity and inherent overfitting mitigation via stochastic multi-sample feature fusion, particularly advantageous for tasks requiring nuanced texture discrimination.

### 4.3. Experimental Environment

The experiment was conducted on a unified hardware setup, including a 12th Gen Intel^®^ Core™ i5-12490F CPU @ 3.00 GHz (Intel Corporation, Ho Chi Minh City, Vietnam) and an NVIDIA GeForce RTX 4060 GPU 8 GB (NVIDIA Corporation, Santa Clara, CA, USA). The operating system was Windows 11 Professional, with Python 3.8.19 as the programming language, PyTorch 1.12 as the deep learning framework, and CUDA 11.3 for GPU acceleration. This setup not only met the requirements for lithium mineral dataset processing but also supported efficient model training and comparative tasks, providing a stable foundation for subsequent experiments.

### 4.4. Evaluation Criteria

To evaluate the model’s overall performance, we used precision (P), recall (R), average precision (AP), and mean average precision (mAP) to measure accuracy across all classes. Parameters (Params), Model Size (Size), Floating-point operations (FLOPs), and Frames Per Second (FPS) were used to assess complexity and real-time performance.

Precision (P) measures the accuracy of positive predictions, while recall (R) measures the proportion of actual positives correctly identified. The formulas are as follows:(19)$$P=\frac{TP}{TP+FP}\times 100\%$$(20)$$R=\frac{TP}{TP+FN}\times 100\%$$
True Positives (TP) are correctly predicted positives, False Positives (FP) are incorrect predictions, and False Negatives (FN) are missed actual positives.

The precision-recall curve is used to calculate the area under the curve (AUC), representing the model’s AP at different recall levels. mAP50 is the average AP at an IoU threshold of 0.50, providing an overall performance evaluation across all classes. The formulas for AP and mAP are as follows:(21)$$AP=\int_{0}^{1}\;P(R)dR\times 100\%$$(22)$$mAP=\frac{\sum_{i=1}^{n}AP_{i}}{n}\times 100\%$$
In this study, n equals 3 due to the inclusion of three mineral categories.

Parameters (Params) represents the model’s complexity, with more parameters requiring greater computational resources. Model size (Size) affects storage and the efficiency of edge deployment. Floating-point operations (FLOPs) measure the computational cost and are linked to the computational power required by edge devices. Frames Per Second (FPS) indicate the inference speed, which is crucial for real-time tasks such as mineral identification.

## 5. Experiments and Discussion

### 5.1. Model Training

The Minima-YOLO architecture was trained on the lithium mineral dataset alongside YOLOv8n and YOLOv8s as baseline comparators under standardized hyperparameters to ensure experimental validity. The training configuration included 200 epochs (batch size = 8), SGD optimization (momentum = 0.937, initial learning rate = 0.01, and weight decay = 5 × $10^{-4}$), and early stopping (patience = 50 epochs). Mosaic augmentation was deactivated during the final 10 epochs to stabilize convergence. Figure 14 details the training dynamics, demonstrating progressive loss reduction and metric stabilization across all models.

As shown in the curves, YOLOv8n and YOLOv8s exhibit significant accuracy fluctuations in the early training stages, failing to improve steadily. This suggests that their structures struggle to quickly adapt to the dataset, leading to instability during training. Additionally, both models show accuracy drops in the mid-training stage, indicating poor adaptability to the dataset and slower convergence, requiring more epochs to achieve better results. In contrast, Minima-YOLO performs exceptionally well in the early training stages, with steadily increasing accuracy and minimal fluctuations. This indicates its ability to quickly extract useful features and adapt to the training data. In the mid-training stage, Minima-YOLO demonstrates a more stable and sustained performance improvements compared to the large fluctuations of the baseline models. The same trends are also observed in the classification loss curve. Notably, Minima-YOLO converges at around 150 epochs, ending training early and being the first among the three models to meet the early stopping condition. This demonstrates that the improved model structure is more efficient, achieving better performance in less time and significantly improving training efficiency. Overall, Minima-YOLO shows significantly better performance during training compared to YOLOv8n and YOLOv8s.

### 5.2. Model Test

The Minima-YOLO model was evaluated on the test set, demonstrating a lightweight architecture with 250 layers, 0.72 million parameters, 2.3 GFLOPs (floating-point operations), and a model size of 1.63 MB. Operating at 103 FPS (Frames Per Second), the model achieved an mAP50 of 99.4%, with perfect precision (100%) for feldspar classification. As detailed in Table 3, minor misclassification between quartz and lepidolite was observed due to their analogous visual characteristics. These results preliminarily demonstrate the overall effectiveness of the proposed improved structure.

### 5.3. Ablation Experiments

To systematically validate the efficacy of individual architectural modifications, YOLOv8-tiny was employed as the baseline model, with proposed enhancements incrementally integrated through a phased implementation protocol. Adhering to the principle of controlled variables, identical hyperparameter settings were applied, and multiple ablation experiments were designed based on mathematical combinations, including the following improvements:A: Replacing the C2f module in the backbone with the Faster-EMA module;B: Replacing the downsampling operation in the backbone with GhostConv;C: Improving the Neck component with the Slim-Neck structure.

The experimental results are summarized in Table 4. The observed high accuracy metrics are primarily attributable to the standardized microscopic imaging protocol and rigorous preprocessing methodologies (segmentation and cropping) that effectively eliminated background interference and minimized False Negatives. This consistency, however, does not preclude comparative model evaluations or architectural optimizations predicated on these performance benchmarks.

Specifically, compared to YOLOv8n, YOLOv8-tiny reduced parameters, model size, and FLOPs by 34%, 33%, and 31%, respectively, while improving FPS by 14%. This demonstrates that limiting and reducing backbone channels can effectively simplify the model structure. Furthermore, applying improvements A, B, and C individually to YOLOv8-tiny showed varying degrees of lightweight optimization. Improvement A, utilizing the Faster-EMA module with PConv and EMA, further lightened the model while slightly improving P, R, and mAP50. However, the deeper network layers introduced by the Faster-EMA Bottleneck led to a decrease in FPS. Combining Faster-EMA with GhostConv (A + B) improved downsampling efficiency through cheap operations, mitigating the FPS drop and further reducing model size and complexity. On the other hand, combining with Slim-Neck (A + C and B + C), produced significant lightweight effects, particularly in reducing parameters and model size.

After incorporating all of the enhancements, the Minima-YOLO model achieved exceptional lightweight performance while maintaining high accuracy. Compared to YOLOv8n, parameters and model size were reduced by 76% and 73%, FLOPs decreased by 72%, and FPS slightly increased by 5%. The ablation experiments validated the effectiveness of these improvements in model lightweighting.

### 5.4. Comparison Experiments

To rigorously validate Minima-YOLO’s performance superiority, comparative analyses were conducted against other YOLO lightweight models and subsequent iterations (v9 [38], v10 [39], v11 [40]). To ensure fairness, all models were trained without pre-trained weights and with their respective default hyperparameter settings, without any optimization or adjustments. The detailed comparison results are shown in the Table 5 and Figure 15.

The comparison results can be grouped into three categories. First, compared to earlier models, Minima-YOLO demonstrated superior performance in both accuracy and complexity, particularly excelling in parameters, model size, and FLOPs. Compared to YOLOv3-tiny, YOLOv5n, YOLOv5s, YOLOv6 [41], and YOLOv7-tiny [42], Minima-YOLO reduced the parameters by 94%, 59%, 90%, 83%, and 88%, respectively, and the model size was reduced by 93%, 56%, 88%, 80%, and 86%, respectively. Additionally, its FPS was only 17% lower than the fastest YOLOv5n. Second, compared to contemporary lightweight models like YOLOv8s and YOLOv8-FasterNet [43], Minima-YOLO maintained its lead in parameters and model size while also improving speed. It reduced FLOPs by 92% and 57%, respectively, and the FPS was 21% higher than YOLOv8s. Although the YOLOv10 and YOLOv11 series offered more efficient structures and slightly outperform Minima-YOLO in FPS, Minima-YOLO still excelled in other lightweight metrics. Its parameters are about one-fourth and its model size one-third of YOLOv10n and YOLOv11n. Notably, compared to YOLOv11s, Minima-YOLO sacrificed only 0.1% mAP50 while reducing the parameters, model size, and FLOPs by 92%, 91%, and 89%, respectively, highlighting its advantages in lightweight design.

Furthermore, comparative analyses were conducted between Minima-YOLO and established object detection architectures, including Faster R-CNN [44], RetinaNet [45], SSD [46], and EfficientDet [47]. The dataset annotations were standardized to comply with each architecture’s specifications, employing default input resolutions during training. The results are presented in Table 6.

Moreover, RetinaNet’s FPS was only 31 frames, indicating poor real-time performance, making it unsuitable for scenarios with high real-time identification requirements. A similar trend was observed in Faster R-CNN and SSD, which not only lagged behind Minima-YOLO in accuracy but also faced challenges in deployment due to their large model sizes. EfficientDet achieved a balance between accuracy and lightweight design but still underperformed Minima-YOLO across all metrics. Figure 16 visually illustrates the differences in FLOPs, parameters, and model sizes across these networks.

### 5.5. Visualization Experiments

Following comprehensive ablation studies and comparative analyses, performance improvements were quantitatively validated. Subsequent evaluations focused on practical applicability through visual comparative assessments. Six representative mineral samples (#1–6) were selected from the validation set, encompassing diverse morphological characteristics across categories (shape, scale, and edge definition).

Benchmark models (YOLOv8n, YOLOv8s, and YOLOv11s) and optimized variants (YOLOv8-tiny and Minima-YOLO) were applied for detection analysis. The comparative visualization results (Figure 17) demonstrated detection consistency across models, with mineral classifications color-coded as follows: red (feldspar), orange-red (quartz), and orange (lepidolite). The confidence scores (0–1 range) quantify detection certainty, where elevated values indicate higher probabilistic assurance of target mineral presence within localized regions.

The results show that all models correctly identified the specific categories of mineral blocks but exhibited differences in confidence scores. Specifically, Block #1 is a feldspar captured under strong lighting. Its reflective surface may resemble quartz, resulting in lower confidence scores for YOLOv8n and YOLOv8-tiny, at 0.59 and 0.65, respectively. In contrast, YOLOv8s, YOLOv11s, and Minima-YOLO achieved confidence scores above 0.9, demonstrating superior distinguishing ability. Block #2 is a feldspar captured under low lighting, with its opaque features well learned by the models. All models showed consistent confidence scores around 0.8. Block #3 is a stacked quartz with thick edges and a glass-like luster. All models correctly identified it with high confidence, with YOLOv11s scoring the highest at 0.97, followed by YOLOv8s and Minima-YOLO at 0.96. Block #4 is a quartz located at the edge of the image. Despite being cropped at the top, its distinct features allowed all models to achieve high confidence scores, with YOLOv11s scoring the highest and YOLOv8n the lowest. Block #5 is a single lepidolite block, appearing transparent against a black background. Minima-YOLO achieved confidence scores comparable to YOLOv8s, both higher than YOLOv8n. Block #6 is a stacked lepidolite with cropped edges. Minima-YOLO still demonstrated a high confidence level comparable to YOLOv8s, and similarly higher than YOLOv8n. Minima-YOLO demonstrated superior performance in microscopic mineral identification, maintaining competitive accuracy with YOLOv8n/YOLOv8s. The optimizations achieved an effective balance between computational efficiency and detection fidelity, establishing a robust solution for resource-constrained mineralogical analysis.

Block #1 (displaying pronounced confidence variance) was selected from the visualization results to investigate hierarchical feature abstraction efficacy. Grad-CAM visualization [48] was employed, extracting fourth-layer backbone feature activations from both architectures. Comparative visualization revealed differential attention allocation patterns between YOLOv8n and Minima-YOLO, with the latter demonstrating an enhanced focus on texturally discriminative mineralogical signatures. The results are shown in Figure 18.

It can be observed that the extracted features vary significantly across different feature maps, with some focusing on edges and others emphasizing the center. From the overall results, Mnima-YOLO more clearly highlighted the actual texture and edge details of Block #1, which is also reflected in its higher confidence scores.

This difference arises from the distinct ways the two models process input images. YOLOv8n uses traditional convolution for downsampling and the C2f module for feature extraction, whereas Minima-YOLO employs two rounds of GhostConv operations and Faster-EMA modules. More specifically, compared to the CBS module’s traditional convolution that generates redundant feature maps, GhostConv achieves downsampling more efficiently and with lower cost through cheap operations, ensuring a lightweight design. Additionally, the Faster-EMA Bottleneck structure within the Faster-EMA module enables Minima-YOLO to use PConv for reduced computational cost while introducing the EMA attention mechanism. This redesigns the multi-branch parallel structure to effectively capture cross-channel and spatial interactions, allowing the module to extract both local details and global features, achieving a comprehensive understanding of the input data. These synergistic architectural optimizations collectively enable Minima-YOLO to achieve feature discriminability comparable to YOLOv8n while attaining enhanced granularity in textural feature capture for specific mineral particulates, thereby elucidating the observed confidence score differentials in localized detection tasks.

### 5.6. Generalization Experiments

Minima-YOLO was tested on a public rock dataset [49] (1180 images, including common rock types, such as limestone, sandstone, and mudstone) to validate its lightweight design and generalization. As detailed in Table 7, Minima-YOLO achieved comparable detection accuracy (mAP50 variance < 2.8%) to the baseline models while significantly reducing parameters by 76.1% (0.72M vs. 3.01M) and FLOPs by 71.6% (2.3G vs. 8.1G), maintaining a leading performance across all lightweight evaluation metrics. Additionally, the visualization results in Figure 19 demonstrate a detection performance comparable to that of YOLOv8n, highlighting both efficient feature transferability and cross-domain potential for mineral detection.

## 6. Conclusions

Traditional lithium mineral identification methods rely on manual approaches, which suffer from inefficiency, misclassification, missed detections, and contamination from chemical reagents. With advancements in computer vision, non-contact detection techniques based on machine vision for mineral identification and classification have emerged. However, these methods still face challenges in edge device deployment and rapid identification.

To address these challenges, we proposed Minima-YOLO, a lightweight YOLOv8-based method for visible light microscopy identification of lithium mineral compositions. The key contributions include the following:**Novel Dataset Construction.** Industrial-grade milled/dried lithium powders were sampled from production lines, and high-resolution microscopy images were acquired and processed via novel segmentation/cropping techniques. The images were annotated and augmented to curate a specialized lithium mineral block dataset.**Novel Network Construction.** The YOLOv8-tiny network was initially designed by simplifying the backbone through adjustments to the number of channels and imposing a limit on the maximum channel count. Subsequently, the Faster-EMA Bottleneck was introduced as the core structure to develop the Faster-EMA feature extraction module, replacing the original C2f module. Additionally, standard convolutions were replaced with the cost-efficient GhostConv for downsampling. Finally, the Slim-Neck structure was incorporated to further enhance the model’s lightweight design.**Extensive Validation Experiments.** Multiple ablation experiments were conducted to validate the effectiveness of the proposed improvements. Subsequently, a comprehensive comparison of various metrics was performed against similar lightweight and other detection models. Finally, visualization experiments were used to highlight the identification performance and generalization capability of the improved model in practical applications.**Excellent Experimental Results.** Minima-YOLO achieved an mAP50 of 99.4%, an FPS of 103, and reduced FLOPs, parameters, and model size to 2.3G, 0.72M, and 1.63MB, respectively. Compared to the baseline YOLOv8n, these metrics were reduced by 72%, 76%, and 73%, with a 5% increase in FPS. This satisfies the demand for high accuracy, rapid identification, and lightweight design, making it highly suitable for industrial applications and deployment on edge devices for lithium mineral detection.**Valuable Reference.** Deep learning was innovatively applied for lithium mineral identification and classification under microscopic conditions. To accommodate this unique environment, the network architecture and modules were specifically designed and improved, offering valuable insights and practical references for the industry.

Moving forward, we plan to further refine this method and explore its applicability to other minerals. We aim to enable rapid classification and purity assessment of finished minerals, providing technical guidance for mining operations. This approach will reduce labor costs, minimize waste generation, improve mining efficiency, and advance intelligent mineral detection and classification.

## Figures and Tables

**Figure 1 sensors-25-02048-f001:**
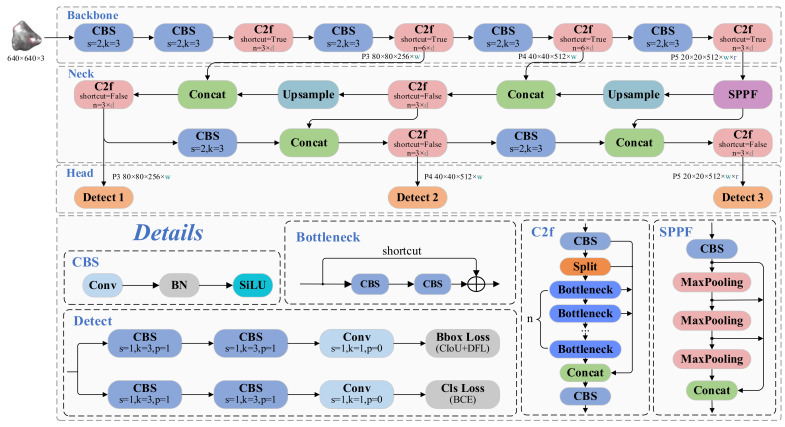
Schematic diagram of the YOLOv8 network architecture. The variables w, d, and r represent the widen factor, deepen factor, and ratio, respectively.

**Figure 2 sensors-25-02048-f002:**
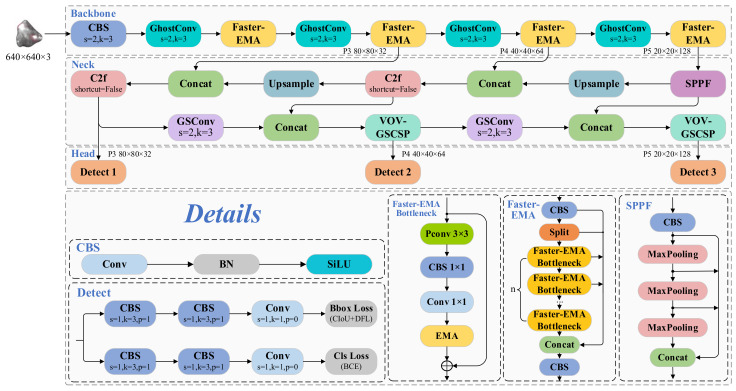
Schematic diagram of the Minima-YOLO network architecture.

**Figure 3 sensors-25-02048-f003:**
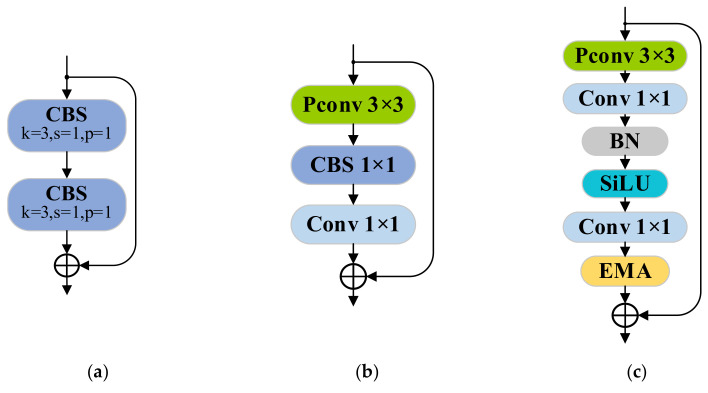
Comparison of bottleneck structure. (**a**) Darknet Bottleneck; (**b**) FasterNet block; (**c**) Faster-EMA Bottleneck.

**Figure 4 sensors-25-02048-f004:**
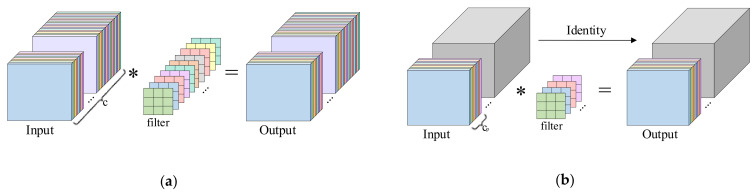
Comparison of standard convolution and partial convolution; “∗” denotes the convolution operation. (**a**) Schematic diagram of standard convolution; (**b**) schematic diagram of partial convolution.

**Figure 5 sensors-25-02048-f005:**
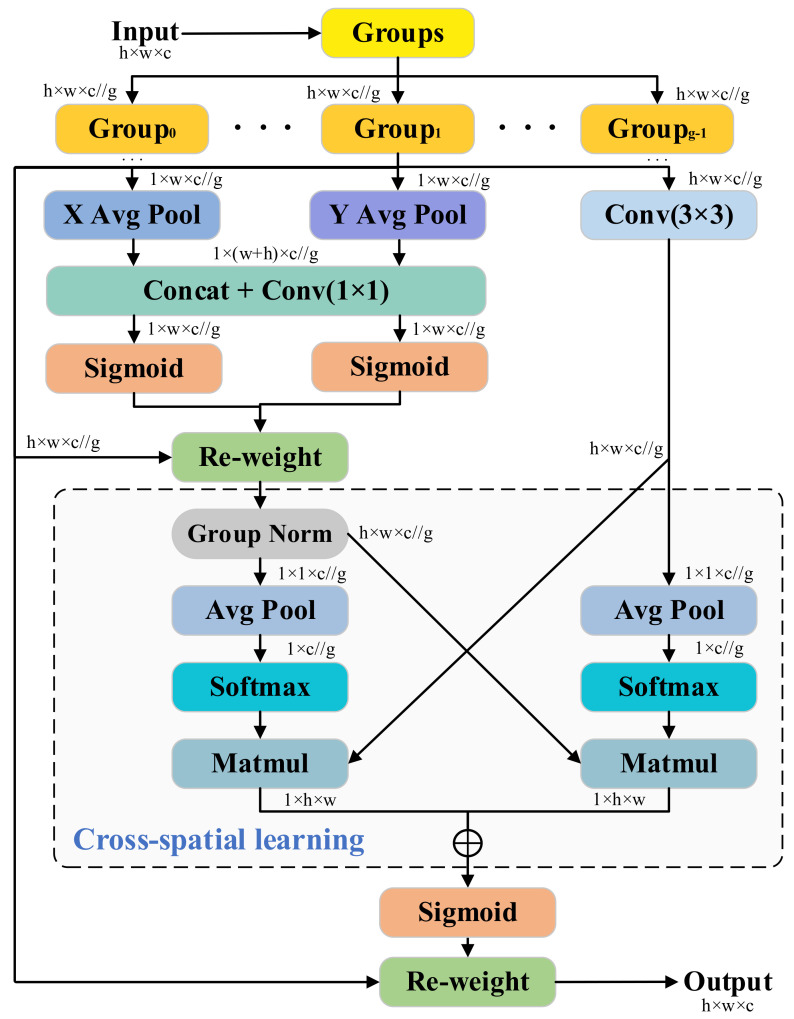
Schematic diagram of EMA module.

**Figure 6 sensors-25-02048-f006:**
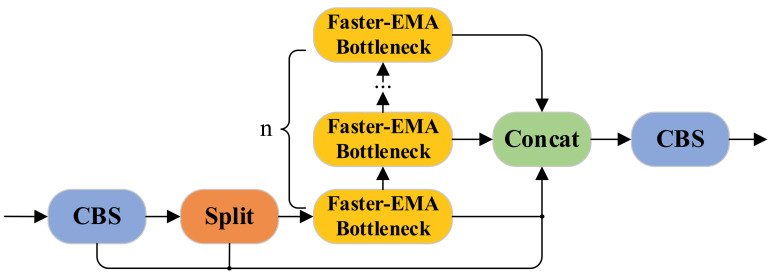
Structure diagram of the Faster-EMA module.

**Figure 7 sensors-25-02048-f007:**
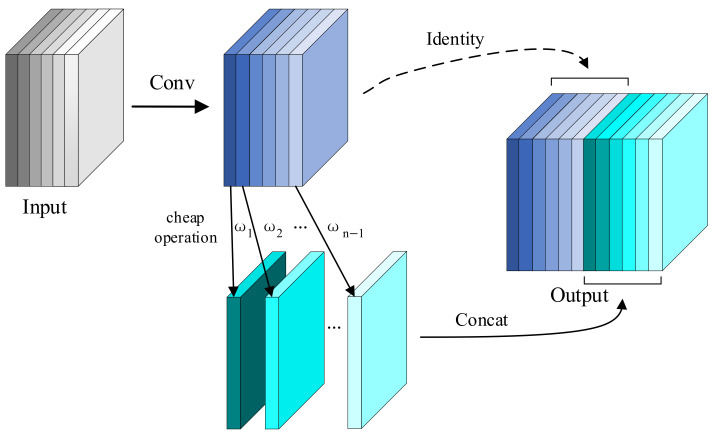
Schematic diagram of GhostConv.

**Figure 8 sensors-25-02048-f008:**
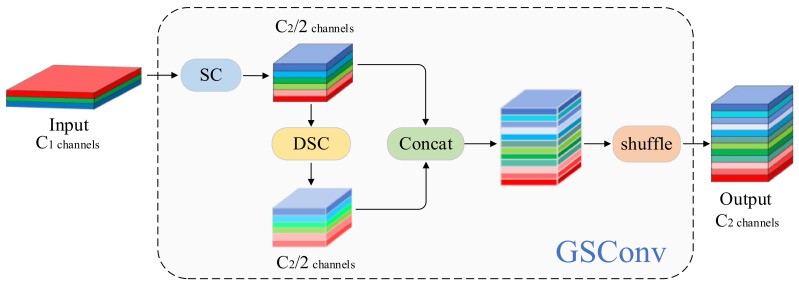
Schematic diagram of GSConv.

**Figure 9 sensors-25-02048-f009:**
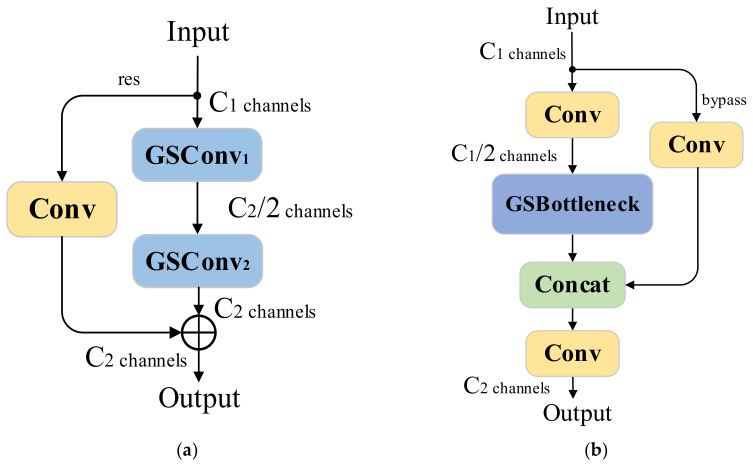
Schematic diagram of the components of Slim-Neck. (**a**) GSBottleneck structure; (**b**) VoV-GSCSP structure.

**Figure 10 sensors-25-02048-f010:**
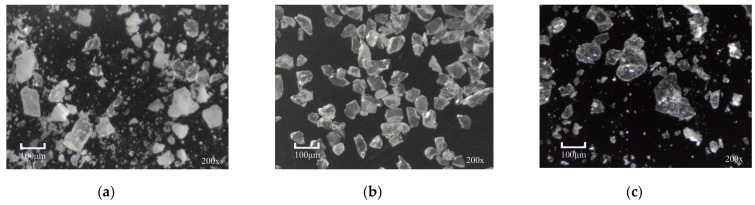
Initial lithium mineral micrographs. (**a**) Feldspar; (**b**) quartz; (**c**) lepidolite.

**Figure 11 sensors-25-02048-f011:**
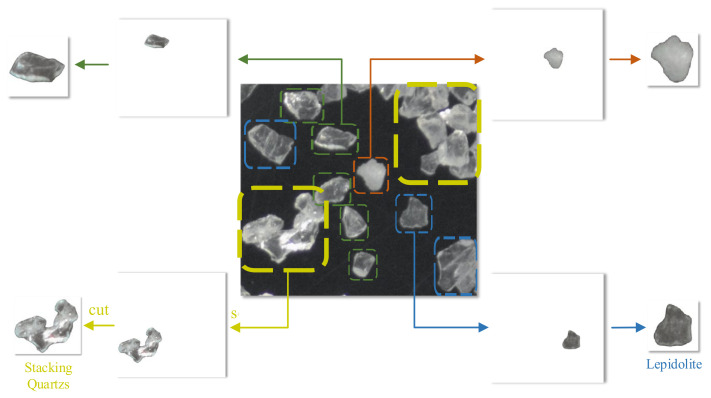
The micrograph processing procedure.

**Figure 12 sensors-25-02048-f012:**
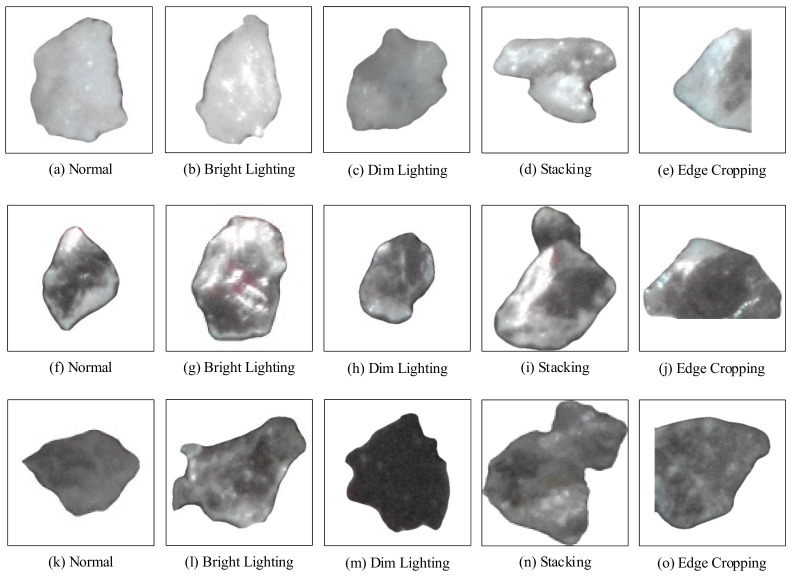
Representative ore block examples under different conditions: feldspar (**top row**), quartz (**middle row**), and lepidolite (**bottom row**).

**Figure 13 sensors-25-02048-f013:**
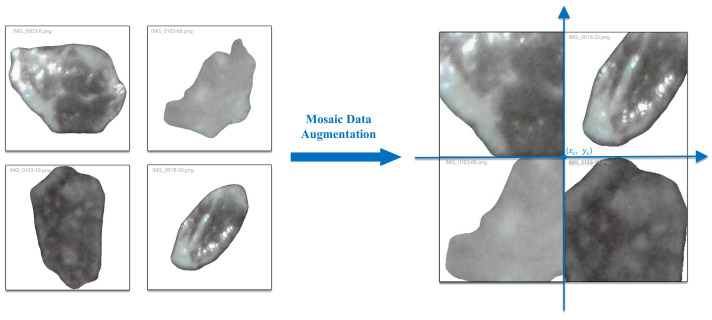
Data augmentation example.

**Figure 14 sensors-25-02048-f014:**
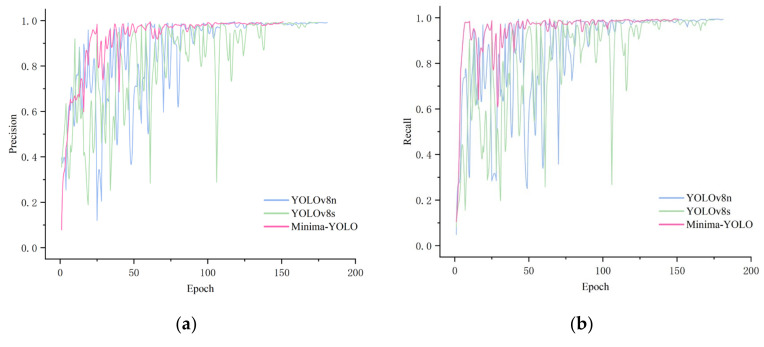
Training curves of YOLOv8n, YOLOv8s, and Minima-YOLO. (**a**) Precision-epoch curve; (**b**) recall-epoch curve; (**c**) mAP50-epoch curve; (**d**) Cls loss-epoch curve.

**Figure 15 sensors-25-02048-f015:**
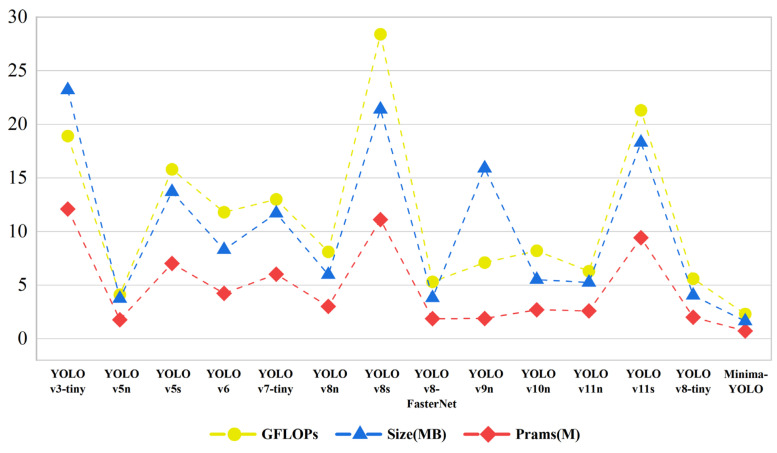
Line chart of lightweight results of comparison with other YOLO models.

**Figure 16 sensors-25-02048-f016:**
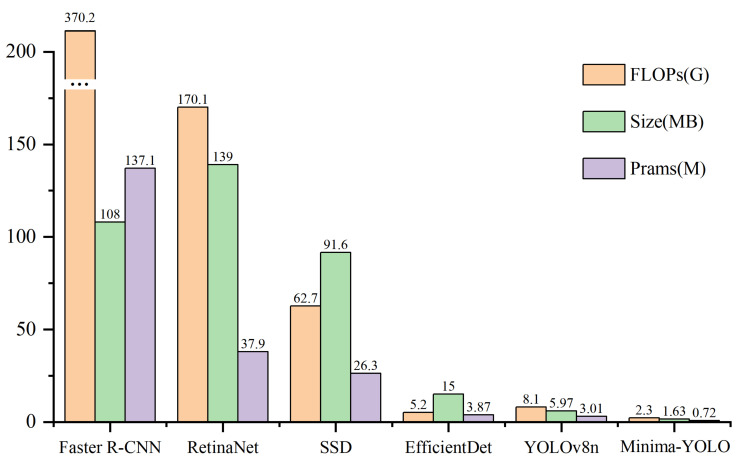
Bar chart of comparison results with other models.

**Figure 17 sensors-25-02048-f017:**
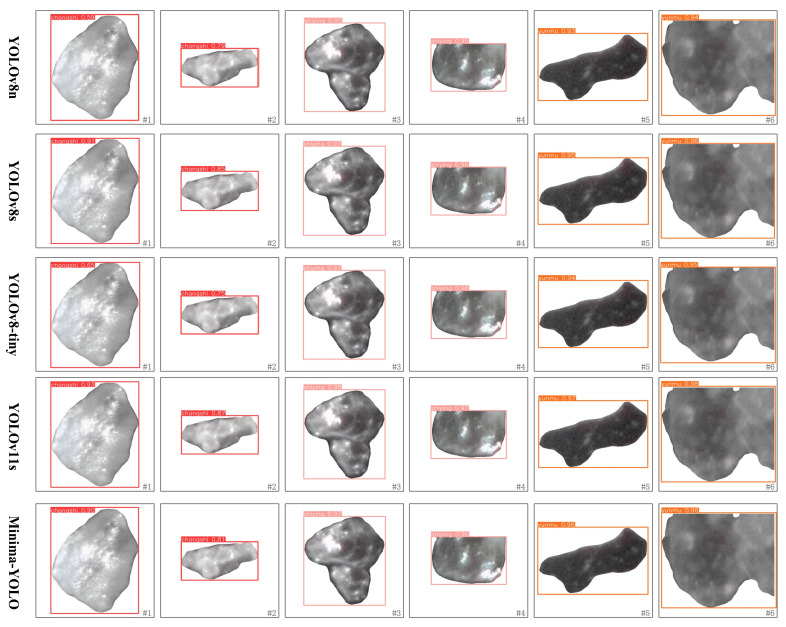
Visualization comparison of identification results.

**Figure 18 sensors-25-02048-f018:**
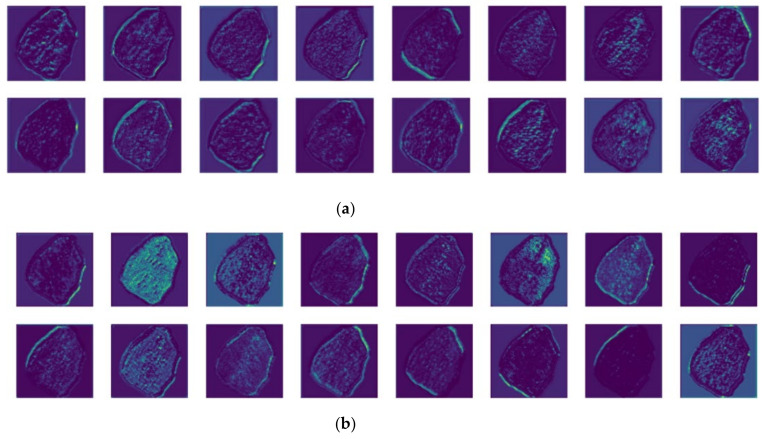
Visualization comparison of feature maps. (**a**) Output feature map of YOLOv8n fourth layer; (**b**) output feature map of Minima-YOLO fourth layer.

**Figure 19 sensors-25-02048-f019:**
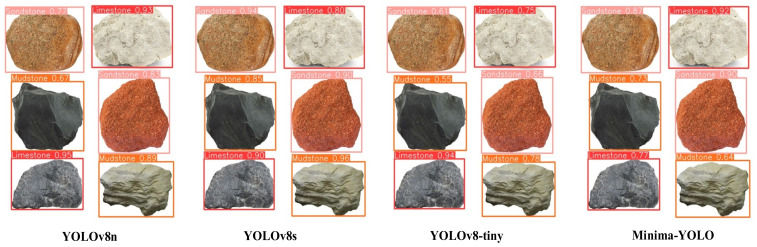
Visualization comparison on other datasets.

**Table 1 sensors-25-02048-t001:** Comparison of channel numbers in YOLOv8 backbones.

Network	0	1	2	3	4	5	6	7	8	9	Max_c	Prams (M)
YOLOv8n	16	32	32	64	64	128	128	256	256	256	1024	3.01
YOLOv8s	32	64	64	128	128	256	256	512	512	512	1024	11.1
YOLOv8-tiny	8	16	16	32	32	64	64	128	128	128	512	1.99

**Table 2 sensors-25-02048-t002:** Details of dataset partitioning.

	Training Set	Validation Set	Test Set	Total
Total	2560	320	321	3201
Feldspar	844	123	104	1071
Quartz	901	101	97	1099
Lepidolite	815	96	120	1031

Note: The sum of each row represents the total number of instances in that category, while each column represents the number of samples within that set.

**Table 3 sensors-25-02048-t003:** Test results of Minima-YOLO.

Class	Instances	P (%)	R (%)	AP50 (%)
All	321	98.9	99.1	99.4
Feldspar	104	1	99.4	99.5
Quartz	97	99.0	97.7	99.3
Lepidolite	120	97.7	1	99.4

**Table 4 sensors-25-02048-t004:** Results of ablation experiments.

Model	A	B	C	P (%) ↑	R (%) ↑	mAP50 (%) ↑	FLOPs (G) ↓	FPS ↑	Prams (M) ↓	Size (MB) ↓
YOLOv8n				98.9	98.8	99.4	8.1	98	3.01	5.97
YOLOv8-tiny				98.3	97.8	99.4	5.6	112	1.99	4.04
Transition Model	√			**99.5**	98.9	**99.5**	5.4	91	1.91	3.88
		√		98.2	98.5	99.2	5.5	111	1.95	3.95
			√	99.1	97.6	99.3	2.7	116	0.87	1.90
	√	√		98.4	99.0	99.3	5.3	95	1.86	3.80
	√		√	98.9	99.2	**99.5**	2.4	99	0.78	1.75
		√	√	98.6	98.1	99.3	2.6	**117**	0.82	1.82
Minima-YOLO	√	√	√	98.9	**99.1**	99.4	**2.3**	103	**0.72**	**1.63**

Note: “↑” denotes a preference for larger values, while “↓” denotes a preference for smaller values. The best results are highlighted in bold.

**Table 5 sensors-25-02048-t005:** Comparison results with other YOLO lightweight models.

Model	P (%) ↑	R (%) ↑	mAP50 (%) ↑	FLOPs (G) ↓	FPS ↑	Prams (M) ↓	Size (MB) ↓
YOLOv3-tiny	96.2	97.4	98.9	18.9	94	12.1	23.2
YOLOv5n	98.7	98.9	99.0	4.1	125	1.76	3.74
YOLOv5s	99.1	99.4	99.1	15.8	121	7.02	13.7
YOLOv6	98.7	98.5	99.3	11.8	124	4.23	8.29
YOLOv7-tiny	98.1	98.8	99.4	13.0	56	6.01	11.7
YOLOv8n	98.9	98.8	99.4	8.1	98	3.01	5.97
YOLOv8s	98.4	98.4	**99.5**	28.4	85	11.1	21.4
YOLOv8-FasterNet	98.8	98.5	99.4	5.3	105	1.86	3.80
YOLOv9n	96.3	99.4	99.3	7.1	59	1.88	15.9
YOLOv10n	99.0	99.3	99.3	8.2	117	2.69	5.50
YOLOv11n	98.5	99.2	99.4	6.3	**131**	2.58	5.24
YOLOv11s	**99.3**	**99.4**	**99.5**	21.3	126	9.41	18.3
YOLOv8-tiny	98.3	97.8	99.4	5.6	112	1.99	4.04
Minima-YOLO	98.9	99.1	99.4	**2.3**	103	**0.72**	**1.63**

**Table 7 sensors-25-02048-t007:** Comparison results on other datasets.

Model	P (%) ↑	R (%) ↑	mAP50 (%) ↑	FLOPs (G) ↓	FPS ↑	Prams (M) ↓	Size (MB)↓
YOLOv8n	59.9	64.2	64.0	8.1	142	3.01	5.98
YOLOv8s	**67.4**	**65.4**	**65.9**	28.4	131	11.1	21.4
YOLOv8-tiny	53.3	63.7	62.7	5.6	150	1.99	4.04
Minima-YOLO	60.3	61.2	63.1	**2.3**	**159**	**0.72**	**1.64**

**Table 6 sensors-25-02048-t006:** Comparison results with other detection models.

Model	Input Size	P (%) ↑	R (%) ↑	mAP50 (%) ↑	FLOPs (G) ↓	FPS ↑	Prams (M) ↓	Size (MB) ↓
Faster R-CNN	600 × 600	88.1	93.5	95.6	370.2	72	137.1	108
RetinaNet	600 × 600	**99.4**	**99.8**	**99.8**	170.1	31	37.9	139
SSD	300 × 300	97.7	99.0	99.1	62.7	95	26.3	91.6
EfficientDet	896 × 896	99.3	99.3	99.1	5.2	80	3.87	15.0
YOLOv8n	640 × 640	98.9	98.8	99.4	8.1	98	3.01	5.97
Minima-YOLO	640 × 640	98.9	99.1	99.4	**2.3**	**103**	**0.72**	**1.63**

## Data Availability

The data presented in this study are available on request from the corresponding author. The data are not publicly available due to laboratory policies and confidentiality agreements.

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
