# Peer review of "Minima-YOLO: A Lightweight Identification Method for Lithium Mineral Components Under a Microscope Based on YOLOv8"

_sensors, 2025, doi:10.3390/s25072048_

Round 1

Reviewer 1 Report

Comments and Suggestions for Authors

As the authors note, lepidolite is the most common lithium mineral and a key raw material for lithium extraction. Lithium, as is known, is a crucial element in the development of batteries and other high-tech devices. Based on this, the ultimate goal of the work, which is related to increasing lithium extraction efficiency, is highly relevant.

In the extraction of minerals (minerals, gemstones), there are several stages of enrichment, and the work describes the final stage, when samples that have already been separated from the host rock undergo sorting. Generally, during the extraction of valuable minerals, earlier stages utilize X-ray methods based on X-ray absorption, dual-energy processing, segmentation, and classification using neural networks, including Yolo. At these stages, it is important not to discard valuable minerals into the waste pile. This should be emphasized.

Questions and comments:

  1. Micro-Yolo is already mentioned in articles. For example, "Micro-YOLO: Exploring Efficient Methods to Compress CNN based Object Detection Model." 2021. You have a different approach; it may be necessary to specify the distinction to avoid confusion with similar titles.
  2. There are many instances where there is no space at the beginning of a sentence: 56. “… to overcome.For instance …”; 66 “…prior expertise.These methods…”. .....
  3. 126 “Figure Error! Reference source 126 not found.” 160 “Table Error! Reference source not found.”
  4. It would be preferable to indicate the scale on Fig.1.
  5. When describing metrics, it is necessary to explain some abbreviations, for example, “Prams.”
  6. In Figure 15, the vertical axis shows both absolute values and percentages, which is incorrect. If possible, break it into different graphs or provide less information.
  7. Can’t your improved model (Micro-Yolo) be applied with equal success to other datasets? For example, in mineral and gemstone extraction, such as beryls, diamonds, or even in other areas like sorting vegetables and fruits? Just on a different scale.

The gain in processing speed with Micro-Yolo is noticeable. Overall, it would be interesting to test your improved version of Micro-Yolo on a larger dataset, if possible in real conditions.

I would recommend the article for publication after revisions.

Author Response

Thank you for your review. We have carefully prepared responses to your professional comments. Please review the attachment.

Reviewer 2 Report

Comments and Suggestions for Authors

The innovation of the article is weak and the quality is biased.

The comments are as follows,

  1. The paper has formatting errors in the references.
  2. The author should reduce the description of the dataset and move it to the experimental section.
  3. The author should explain the significance of section 2.3 in the paper.
  4. The writing style lacks professionalism. The paper reads more like a "technical blog" than a "research report". Section 3 should focus more on "innovation" and reduce other descriptions.
  5. Figure 6 shows each Faster EMA module. Figure 10 shows that each Faster EMA module consists of n (the specific number is not mentioned) Faster-EMA Bottlenecks. Figure 9 shows the composition of EMA in each Faster-EMA Bottleneck. Each EMA has three branches and involves an attention mechanism. Therefore, I think each Faster EMA module requires a lot of computation and cannot guarantee model lightweightness.
  6. Section 3 has poor organization and needs restructuring. Ghost Convolution belongs to Slim-Neck, so why does it have a separate section? Why don't GSConv and VoV-GSCSP have their own sections? Also, Section 3 has too many section numbers and should be simplified. The "Evaluation Criteria" section should be in the experimental part.
  7. Some recent relevant works should be discussed, such as "An efficient lightweight convolutional neural network for industrial surface defect detection ", "Adaptive critic design for safety-optimal FTC of unknown nonlinear systems with asymmetric constrained-input" and "A novel deep convolutional neural network algorithm for surface defect detection", which may  enhance the credibility  and the impact of the manuscript.
  8. In Figure 17, the precision of the first detection image for YOLOv8n is only 0.59, which doesn't make sense. In fact, any model that innovates based on another model may have differences, but not this significant, especially when the baseline model itself has high precision."
Comments on the Quality of English Language

The quality deviation of English affects understanding

Author Response

(The authors gave the same response as above.)

Round 2

Reviewer 1 Report

Comments and Suggestions for Authors

I believe that the article can be published in its current form. The authors have addressed my comments.

Reviewer 2 Report

Comments and Suggestions for Authors

This version demonstrates significant improvements compared to the previous one. The authors have carefully addressed all of the comments and suggestions, making the necessary revisions to enhance the overall quality of the work.
The manuscript can be accepted.